# Dietary and Physical Activity Habits as Conditioning Factors of Nutritional Status among Children of GENYAL Study

**DOI:** 10.3390/ijerph20010866

**Published:** 2023-01-03

**Authors:** Helena Marcos-Pasero, Elena Aguilar-Aguilar, Gonzalo Colmenarejo, Ana Ramírez de Molina, Guillermo Reglero, Viviana Loria-Kohen

**Affiliations:** 1Nutrition and Clinical Trials Unit, GENYAL Platform, IMDEA-Food Institute, CEI UAM + CSIC, 28049 Madrid, Spain; 2Bioactivity and Nutritional Immunology Group (BIOINUT), Faculty of Health Sciences, Universidad Internacional de Valencia-VIU, Pintor Sorolla 21, 46002 Valencia, Spain; 3Department of Nursing and Nutrition, Faculty of Biomedical and Health Sciences, Universidad Europea de Madrid, Villaviciosa de Odón, 28670 Madrid, Spain; 4Biostatistics and Bioinformatics Unit, IMDEA-Food Institute, CEI UAM + CSIC, 28049 Madrid, Spain; 5Molecular Oncology and Nutritional Genomics of Cancer, IMDEA-Food Institute, CEI UAM + CSIC, 28049 Madrid, Spain; 6Production and Development of Foods for Health, IMDEA-Food Institute, CEI UAM + CSIC, 28049 Madrid, Spain; 7Department of Production and Characterization of Novel Foods, Institute of Food Science Research (CIAL), CEI UAM + CSIC, 28049 Madrid, Spain; 8Grupo de Investigación VALORNUT-UCM, Departamento de Nutrición y Ciencia de los Alimentos, Facultad de Farmacia, Universidad Complutense de Madrid, 28040 Madrid, Spain

**Keywords:** diet, nutrition, nutritional status, children, lifestyle, obesity, physical activity

## Abstract

Excess weight (EW) in children has become a severe public health problem. The present study aimed to describe the main lifestyle characteristics and their possible association with nutritional status in a group of schoolchildren enrolled in the GENYAL study, where 221 children in the first or second grade of primary education (6–9 years old) were included. Anthropometric (BMI and bioimpedance), dietary intake (twice-repeated 24 h food record), and physical activity (twice-repeated 24 h physical activity questionnaire) data were collected. Logistic and linear regressions, with *p*-values adjusted for multiple tests by Bonferroni’s method and with sex and age as covariates, were applied. The prevalence of EW was 19%, 25.4%, and 32.2%, according to Orbegozo Foundation, IOFT, and WHO criteria, respectively. The results showed a significant association between schoolchildren’s nutritional status and energy balance, defined as the ratio of estimated energy intake to estimated energy expenditure (%), (β = −1.49 (−1.9–1.07), *p* < 0.01) and KIDMED Mediterranean Diet Quality Index score (β = −0.19 (95% IC −0.38–0), *p* = 0.04), and between the availability of TV or other technological devices in their room and the child’s BMI (β = 1.15 (95% IC 0.20–2.10), *p* = 0.017) and their fat mass (β = 3.28 (95% IC 0.69–5.87), *p* = 0.013). The number of dairy servings/day had a protective effect against EW (OR = 0.48 (0.29–0.75), *p* adjusted = 0.05)). Studying lifestyle factors associated with obesity is essential for developing tools and strategies for obesity prevention in children.

## 1. Introduction

In line with the WHO European Childhood Obesity Surveillance Initiative, one in three European children aged six to nine were overweight or obese in 2015 [1]. This fact confirms that pediatric excess body weight has become a significant public health problem worldwide.

Notwithstanding the unexpected plateau in childhood obesity rates observed in developed countries [2], Spain continues to have one of the highest European rates [3], as some national researches reveal, such as the ALADINO study (2019) that estimated a rate of childhood overweight prevalence of 23.3% (24.7% girls, 21.9% boys) and an obesity rate of 17.3% (15.0% girls, 19.4% boys) [4].

Cardiovascular diseases, type 2 diabetes mellitus, dyslipidemia, hypertension, and non-alcoholic fatty liver disease, which are associated with premature adult mortality, are non-communicable diseases linked to obesity and their most frequent comorbidities [5]. The severity of obesity is directly related to the risk of cardio-metabolic diseases, particularly among children [6]. Regardless of the multifactorial obesity aetiology, genetic susceptibility plays a specific role. It is dramatically influenced by a permissive and obesogenic environment that begins in the mother’s womb and continues throughout childhood and adolescence [6]. It has been widely known that dietary and physical activity habits are essential health determinants [7] and can act as critical modulators in the prevention and treatment of obesity [8]. Hence, they are essential in the search for tools that might contribute to obesity prevention at an early age, improving the expectations of quality of life and longevity.

Dietary patterns in Spanish children have been getting worse in recent decades. Food consumption has approached westernized countries characterized by an excessive caloric intake, poor nutritional quality food, and low adherence to the typical Mediterranean diet (MD) [9]. Likewise, it has been observed that a high rate of 9–12-year-old Spanish children does not achieve the daily recommendation of practising at least 60 min of moderate- to vigorous-intensity physical activity [10].

Many studies have been conducted worldwide on the association between nutritional and physical activity factors and excess weight in schoolchildren, underlining the need for more research in this area [11,12,13,14,15]. Nevertheless, in Spain, only several studies were conducted, such as The ALADINO study [4] and the EsNuPi study [16], the two most complete national studies on the prevalence of pediatric obesity, but they were descriptive and cross-sectional studies. Other national studies, such as The THAO Child Health Program [17] and the POIBC study [18], included nutritional intervention, but were carried out outside the social context of Madrid. The GENYAL study is a cluster-randomized clinical trial with a 5-year follow-up intervention, based on nutritional education, annual anthropometric measurement evaluations, and data collection from questionnaires developed in Madrid (Spain), which will increase knowledge in this area.

Thus, the objective of this study was to describe the main lifestyle characteristics (diet and exercise) and their possible association with nutritional status in a group of schoolchildren (6–9 years old) enrolled in the GENYAL study (Madrid, Spain) for the prevention of childhood obesity.

## 2. Materials and Methods

### 2.1. Study Population and Design

A cross-sectional study was carried out in the context of the project “GENYAL study for the prevention of childhood obesity”. It is an observational and a 5-year intervention study whose main objective is to design and validate a predictive model to identify those children who would benefit most from actions to reduce the risk of obesity and its complications, considering both environmental and genetic factors involved. Given the large number of endpoints and associations analyzed, and the absence of initial guesses for the variability for many of them, it was not possible to conduct a rigorous and univocal estimation of the sample size. Given the available resources, we decided to use the most significant sample possible. This work was approved by the Research Ethics Committee of the IMDEA Food Foundation with PI number: IM024. The methodology used complies with the Declaration of Helsinki (1964) and its modifications. The project has been coded as Clinical Trial (www.clinicaltrials.gov, accessed on 29 October 2018) with the number NCT03419520. This study began in March 2017, and the initial assessment results have been analyzed here.

The Consejería de Educación e Investigación de la Comunidad de Madrid was responsible for selecting six primary schools in the city of Madrid (Spain), two in the north, two in the centre, and two in the south. This selection was based on the number of students per school and the socioeconomic level of the neighbourhoods, achieving a representative selection of the average income of the city households. The sample size calculation could not be defined due to the characteristics and design of this study, where we had no previous data for the variability of the outcomes in the population, with this being a study of exploratory character.

In order to participate, the school-aged children had to be studying the 1st or 2nd grade of primary education, attend the centre on the evaluation days, and not expect a change of school in the forthcoming years. The signed informed consent was collected by at least one of the parents of all participating children.

### 2.2. Anthropometric Data

The evaluations were taken by trained dietitians using a standardized protocol, following the guidelines proposed by the International Society for the Advancement of Kinanthropometry [19]. For all the parameters, an average of 2–3 measurements were made depending on the magnitude of the differences in each record.

The children were evaluated barefoot, with shorts and an undershirt on. Height was measured using a stadiometer with millimetric accuracy (Leicester-Biología Tecnología Médica SL, Barcelona, Spain). Body weight and fat mass (FM) were determined using a bioimpedance with electrodes for hands and feet (BF511, Omron Healthcare Co., Ltd., Kyoto, Japan). Based on these data, other derived variables of interest were calculated. The body mass index (BMI) was calculated as weight (kg)/height (m)^2^ and classified according to percentiles of the Faustino Orbegozo Foundation [20], the International Obesity Task Force (IOTF) [21], and the World Health Organization (WHO) growth standards [22]. 

### 2.3. Dietary Data

To obtain a dietary data compilation, a twice-repeated 24 h food record was completed over two non-consecutive days (one weekday and one weekend day), following the European Food Safety Authority guidelines [23]. The dietary data transformation to macro and micronutrients was done through the DIAL software (Alce Ingeniería, Madrid, España). The energy balance (%EB) was calculated as total energy intake (TEI) (kcal/day)/total energy expended (TEE) (kcal/day) × 100 (where TEI was obtained from the DIAL software, and TEE was estimated from the WHO equation [24] that establishes a basal expenditure based on the weight, age, and sex of each individual and then, this basal expenditure is multiplied by an Individual Physical Activity Coefficient (IPAC)). When the EB > 110%, it was considered overfeeding, between 90–110% remains in equilibrium, and <90% insufficient intake.

Data were also recorded from the children’s food consumption habits since birth and their parents’ habits. The “KIDMED Mediterranean Diet Quality Index” was utilized to evaluate their adhesion to the Mediterranean eating pattern. This questionnaire included 16 dichotomous questions with an affirmative or negative answer, obtaining a final score which ranged from 0 to 12. The questions included were (Q1) takes a fruit or fruit juice every day; (Q2) has a second fruit every day; (Q3) has fresh or cooked vegetables regularly once a day; (Q4) has fresh or cooked vegetables more than once a day; (Q5) consumes fish regularly (at least 2–3 times per week); (Q6) goes more than once a week to a fast-food (hamburger) restaurant; (Q7) likes pulses and eats them more than once a week; (Q8) consumes pasta or rice almost every day (5 or more times per week); (Q9) has cereals or grains (bread, etc.) for breakfast; (Q10) consumes nuts regularly (at least 2–3 times per week); (Q11) uses olive oil at home; (Q12) skips breakfast; (Q13) has a dairy product for breakfast (yoghurt, milk, etc.); (Q14) has commercially baked goods or pastries for breakfast; (Q15) takes two yoghurts and/or some cheese (40 g) daily; and (Q16) takes sweets and candy several times every day. Questions denoting a negative connotation concerning the Mediterranean diet were assigned a value of −1 (questions Q6, Q12, and Q16), and those with a positive aspect +1. The sums of the values from the administered test were classified into three levels: (1) >8, optimal Mediterranean diet; (2) 4–7, improvement needed to adjust intake to Mediterranean patterns; and (3) ≤3, low diet quality [25].

### 2.4. Physical Activity Data

A twice-repeated 24 h physical activity questionnaire (a weekday and a weekend day) was collected [26]. The parents had to specify the time that their children spent during 24 h on a weekday and 24 h on a weekend day doing different activities, including resting hours and activities with a variable level of intensity (very light, light, moderate, and intense). According to the activity coefficient defined by the WHO [27], the time spent doing different activities was multiplied, added, and divided by 24, obtaining the Individual Physical Activity Coefficient (IPAC). Firstly, the weekend day IPAC was multiplied by 2 and the weekday IPAC by 5. Then, the mean physical activity per individual was obtained by adding both results and dividing by 7. An equivalence between the IPAC and the Physical Activity Coefficient (PAC) was made according to sex, following the Institute of Medicine methodology [28]. To conclude, participants were classified for their PAC into very active, active, low active, and sedentary subjects. In order to estimate if children achieved the recommendation to do more than 60 min/day of active and moderate sports activities [29], the total active weekly hours (TAWH) were calculated considering the time invested in vigorous and moderate extra-curricular activities and two additional curricular hours.

### 2.5. Statistical Analyses

Descriptive analyses were performed by computing the class’s absolute and relative frequencies for the categorical variables. The mean, median, standard deviation, interquartile range, maximum, and minimum were calculated for the quantitative variables. The Shapiro–Wilk test (*p* > 0.05) was used to assess the normality of data. To check the homogeneity between groups, a *t*-test was used for quantitative variables with normal distribution and a Mann–Whitney U test in the opposite case. Chi-Square or Fischer exact tests were used for categorical variables. 

The associations between anthropometric, physical activity, and nutritional data were established through logistic and linear regressions/ANOVA adjusted by sex and age. No variable selection approaches were used (e.g., stepwise), and for each association tested, a new regression model was developed with the two involved variables included (one as predictor and the other as outcome) plus the two adjustment variables as additional predictors. The *p*-values were corrected by means of the Bonferroni method. This correction was motivated by computational simplicity and the small number of tests performed. The multiple comparison analysis included a total of 50 variables from different categories included in the study (nutritional, anthropometric, socio-sanitary, etc.). The overweight and obesity categories were unified in some analyses as a single category called excess weight (EW). Statistical analyses between the study variables and nutritional status were performed with BMI as a continuous variable and as a BMI category according to IOTF criteria as a categorical variable, including underweight, average weight, and overweight. Statistical tests were two-tailed with a 5% level of significance. R Statistical Software 3.1 (www.r-projet.org) (accessed on 29 October 2018) was used to perform the statistical analysis.

## 3. Results

A total of 569 schoolchildren from the 6 contacted schools were included, and 39.36% of the parents signed the informed consent form. Among the 224 children included, 3 did not attend the evaluation day, so the final sample consisted of 221 (48.0% girls and 52.0% boys) with a mean age of 6.75 ± 0.73 years. The descriptive anthropometric, physical and leisure activities, and intake characteristics of schoolchildren according to sex are summarized in Table 1.

According to the WHO criteria, 32.2% of the students evaluated had EW (18.1% overweight, 14.1% obesity). These figures were higher than when the IOFT standard (25.4%) or the national criteria of the Orbegozo Foundation (19.0%) were applied. The percentages of mean values of nutrient intake over the total caloric value were 44.48% in carbohydrates, 20.15% simple sugars, 18.16% fibre, 16.55% protein, 38.96% of fats, 13.28% SFA, 17.22% MFA, and 4.84% PFA. Boys were more likely to dedicate weekly to active physical activities than girls.

The main dietary characteristics of schoolchildren according to their nutritional status are summarized in Table 2. Eleven underweight children (five percent) were found in the sample following the IOTF criteria. Given the small sample size, they could not be treated independently.

Regarding schoolchildren’s energy balance (EB), calculated as TEI (kcal/day)/TEE (kcal/day) × 100, 38.8% of the children were overfeeding, while 28.9% were classified as having insufficient intake. When the relationship between the %EB (independent variable) and schoolchildren’s BMI and FM (dependent variables) was evaluated, it was found to be an inverse and significant association (β = −1.49 (−1.9–1.07), *p* = 1.53 × 10^5^ (*p* < 0.01) and β = −3.46 (−4.62–2.29), *p* = 1.24 × 10^8^ (*p* < 0.01), respectively).

Nevertheless, the intake of macronutrients, fibre, simple sugars, and lipid percentage were not associated with the child’s nutritional status.

The number of servings per day for the different food groups considered according to the schoolchildren’s nutritional status is presented in Figure 1. The number of dairy servings/day showed a protective effect against EW (OR = 0.48 (0.29–0.75), *p* adjusted = 0.05).

About 80.0% of the children had 5 or more meals daily, and 90.0% had breakfast regularly. Nonetheless, only 24.0% of the total schoolchildren had a breakfast considered as a well-balanced one (fruits, dairy, and cereal products). Neither the number of meals made per day nor the consumption of a quality breakfast showed an association with EW in schoolchildren.

Concerning the KIDMED questionnaire, more than 50.0% of the children needed to improve their diet and 7.50% had a very low-quality diet. The total score obtained in the KIDMED questionnaire was inversely associated with the schoolchildren’s BMI (β = −0.19 (95% IC −0.38–0), *p* = 0.04) and FM (β = −0.65 (−1.15–0.14), *p* = 0.01). 

Finally, when other factors were considered, such as if the children received breastfeeding, its length of time, or the age of Beikost (defined as an infant’s first non-liquid food) beginning, neither of them was associated with the nutritional status. Table 3 describes the primary schoolchildren’s physical activity characteristics and sleep length according to their nutritional status.

According to the Individual Physical Activity Coefficient (IPAC), 55.6% of the children were classified as sedentary or low-active subjects. Moreover, the coefficient was lower during weekends (1.52 ± 0.13 at the weekend and 1.61 ± 0.12 during the week, *p* < 0.001). At the same time, only 40.0% of the total achieved the daily recommendation of practising at least 60 min of moderate- to vigorous-intensity physical activity [20]. 

In addition, 22.0% of children spend more than two hours of TV time a day (television, playing video games, computer…), mainly during the weekend. Likewise, 15.2% had a TV or other technological devices in their room. Concerning relation to this, a direct and significant association was observed between the availability of these devices and the child’s BMI (β = 1.15 (95% IC 0.20–2.10) *p* = 0.017) and their FM β = 3.28 (95% IC 0.69–5.87) *p* = 0.013. Only the relationship between the %EB and schoolchildren’s BMI (*p* < 0.01) and FM (*p* < 0.01) maintained significance after multiple comparison analyses.

## 4. Discussion

Detecting harmful dietary and sedentary habits could be advantageous in developing effective nutritional and educational intervention strategies towards obesity prevention [8]. The results of the anthropometric evaluation showed that a fourth of the children who participated in the study presented excess weight. These results parallel those obtained in the ALADINO study in 2019 [4].

The child’s energy balance (TEI (kcal/day)/TEE (kcal/day)) showed a paradoxical inverse association with the BMI in the same way described previously [30,31,32]. It has been known that the weight status of children can lead to under- and over-reporting of a child’s dietary and physical activity behaviour by parents. This results from social desirability and sociocultural biases when being overweight is seen as a desirable trait or if the prevalence is high and being overweight is considered “normal” relative to other children [33].

In this work, the diet’s distribution of proteins and fats and the simple sugar contribution were far from the recommendation, and they fell short of total carbohydrates [34]. This has been previously observed in different national children surveys [35] and in the EsNuPi study [16]. These data reflect that the diet in children must be improved. Excess protein in the diet has previously been associated with obesity and other metabolic risks at an early age and higher BMI later in childhood [36]. It maintains the same trend in terms of excess fat [37]. Nevertheless, these aspects of the diet were not associated with the schoolchildren’s nutritional status categories or with the BMI and FM in this study. 

Significant differences were observed between the hours dedicated to practising physical activity weekly (TAWH) between boys and girls, observing that the boys spent the most time practising. These results have been observed in other studies [11,14]. In the ALADINO 2019 study, this same trend was observed. In addition, boys carried out more extracurricular sports activities than girls, reflecting an inequitable distribution between genders in sports since childhood [4].

The intake of calcium from dairy and the presence of peptides and bioactive components with a potential effect on satiety and weight control [38,39] could explain the significant association found between the number of daily portions of dairy consumed and the BMI of the children in the study. According to the latest reviews on the subject, milk and dairy products are not consistently associated or are inversely associated with obesity and indicators of adiposity in children [40]. For this reason, policies that intend to increase dairy consumption could be crucial to preventing “weight excess” in children and adults. Breakfast is often described as the most important meal of the day, and its omission has been related to adiposity among children and adolescents [41]. Although practically all the children had breakfast regularly in this study, only one out of four consumed a well-balanced breakfast. Unlike other studies [15,42], this one did not identify an association between this kind of breakfast consumption and obesity. The different methodologies employed to characterize a healthy breakfast could explain the varied results.

The Mediterranean diet is considered a model of a healthy diet. It is likely because its composition is rich in vegetables, fruits, legumes, whole cereals, and many sources of fibre and antioxidants. The score obtained in the KIDMED questionnaire confirmed the distancing of the Spanish children from the MD pattern described in previous studies [43] and is similar to the results reported by the ALADINO 2019 study [4]. This fact is considered one of the reasons why Spain is among the European countries with the highest prevalence of childhood obesity [44]. However, in the EsNupi study [16], a main dietary pattern was observed related to the Mediterranean diet, with high consumption of cereals, fruits, vegetables, milk, and dairy products. On the plus side, this result showed the association of this diet with a lower prevalence of obesity [45]. 

In this study, 22.0% of children spent more than two hours of TV time a day (television, playing video games, computer…), mainly during the weekend. In the ALADINO 2019 study, the percentage was higher, and the same trend was observed towards greater inactivity and passive leisure on the screen during weekends [4]. The connection observed in this study between the availability of TV and other devices in the child’s room and excess weight could be related to increased meal frequency and food intake or to less time that children are under adult supervision. Unfortunately, these aspects were not covered in this study. Similarly, it could avoid the performing of active leisure time [46,47]. Moreover, in other studies, total screen time has been directly related to excess weight in children [11].

The inverse relationship between sleep length and BMI has been described in children and adolescents. Recent studies associate it with a higher intake and a lower energy expenditure due to the hormonal pathways involved in the appetite for partial sleep deprivation [48]. In this study, the children who presented obesity had less average sleep length; however, the differences were not significant. It was observed that children with obesity performed more physical activity through the calculated variables (IPAC and TAWH) than children with average weight and who were overweight. The results did not obtain statistical significance. In the ALADINO 2019 study, contrary data was obtained. Children with average weight were the most active [4].

We can identify some limitations in this study. Firstly, the use of dietary and physical activity questionnaires has been criticized. Nevertheless, in the absence of better tools with a low cost and high throughput to estimate the intake, consumption records can offer valuable information, although they should be interpreted cautiously. A second limitation is the sample size (*n* = 221). Nevertheless, it is essential to consider that this study is framed as an intervention study of 5 years follow-up. 

On the other hand, the lack of sample size calculation is one of the weak aspects of our study. Due to the nature of the study as a clinical trial, the large number of variables used, and the duration of 5 years, a statistically robust sample size could not be implemented. Furthermore, the “Consejería de Educación e Investigación de la Comunidad de Madrid” was responsible for the selection of six representative schools of the Community of Madrid (CM) (Spain), considering the number of students per centre and the average and socioeconomic level of the districts and neighbourhoods. Therefore, the selection was representative of the average income of the CM households. The lack of other confounding factors, such as sociodemographic variables of the characteristics of the families or genetics variables, is considered a limitation of this study. Nevertheless, in the GENYAL study, these variables were collected and will be evaluated within other studies with framed objectives on this subject.

From a statistical point of view, using a simple conservative Bonferroni correction that controls the family-wise error rate can lead to an increased rate of false negatives compared to other methods. However, given the small number of tests performed, this should not be a significant issue.

By contrast, it is worth highlighting the homogeneity of the population in terms of distribution by sex, as well as its representativeness since six schools from three different areas of the CM participated, which allows us to have a better knowledge of the situation throughout the Community and not from a specific school or area.

## 5. Conclusions

In conclusion, the results obtained in this study reflect that a high prevalence of excess weight was associated with the intake of dairy servings, the distancing of the Mediterranean diet pattern, and the presence of TV or other devices in children’s rooms. The control of these aspects could be an important point to guide families in lifestyle improvement. However, the small sample size of this study requires that its results be corroborated in future studies with larger sample sizes.

## Figures and Tables

**Figure 1 ijerph-20-00866-f001:**
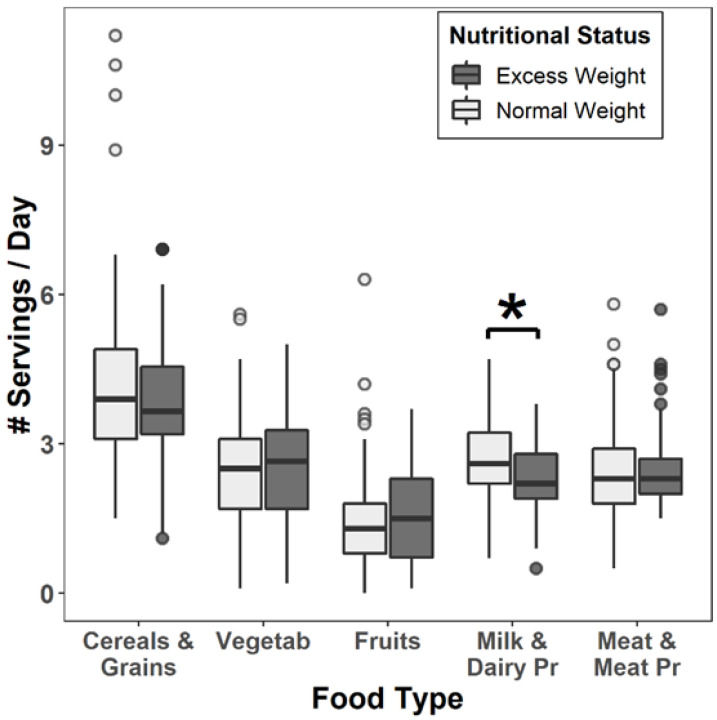
Amount of servings/day for the different food groups according to the schoolchildren’s nutritional status. (* *p* = 0.039 Difference between the number of milk and dairy products servings/day according to nutritional status. Data adjusted by age and sex).

**Table 1 ijerph-20-00866-t001:** Sample characteristics by sex.

	Total		Girls		Boys		
	N	x ± SD	N	x ± SD	N	x ± SD	*p*
Height (cm)	221	124.74 ± 6.41	105	123.75 ± 6.63	116	125.63 ± 6.10	0.029
Weight (kg)	221	26.60 ± 6.03	105	26.37 ± 6.07	116	26.81 ± 6.00	0.555
Fat mass (%)	218	20.59 ± 7.17	103	20.50 ± 7.60	115	20.67 ± 6.80	0.635
BMI (kg/m2)	221	16.92 ± 2.63	105	17.04 ± 2.73	116	16.82 ± 2.55	0.448
IPAC	198	1.58 ± 0.11	92	1.57 ± 0.09	106	1.60 ± 0.12	0.054
Sleeping hours	198	9.92 ± 1.09	92	9.92 ± 1.19	106	9.92 ± 1.00	0.938
TAWH (h)	221	3.74 ± 1.81	105	3.46 ± 1.62	116	4.03 ± 1.94	0.025
TEE (kJ/day)	198	7256.02 ± 1000.69	92	7103.49 ± 975.32	106	7388.40 ± 1008.10	0.029
TEI (kJ/day)	201	7755.46 ± 1407.94	93	7582.32 ± 1286.16	108	7894.91 ± 1494.10	0.125
CHD (% TEI)	201	44.48 ± 5.30	93	44.43 ± 5.73	108	44.53 ± 4.93	0.900
Simple sugars (% TEI)	201	20.15 ± 4.08	93	20.17 ± 3.63	108	20.13 ± 4.45	0.539
Vegetable fibre (g)	201	18.17 ± 5.82	93	17.76 ± 5.73	108	18.52 ± 5.91	0.185
Proteins (% TEI)	201	16.55 ± 2.17	93	16.60 ± 2.16	108	16.51 ± 2.18	0.778
Fats (% TEI)	201	38.96 ± 5.02	93	38.96 ± 5.44	108	38.95 ± 4.65	0.987
SFA (% TEI)	201	13.29 ± 2.27	93	13.25 ± 2.35	108	13.33 ± 2.20	0.806
MFA (% TEI)	201	17.23 ± 3.25	93	17.20 ± 3.49	108	17.25 ± 3.04	0.927
PFA (% TEI)	201	4.84 ± 1.50	93	4.83 ± 1.57	108	4.85 ± 1.44	0.826
Cereals and grains (s/d)	201	4.05 ± 1.45	93	3.95 ± 1.59	108	4.15±1.32	0.069
Vegetables (s/d)	201	2.48 ± 1.05	93	2.51 ± 1.13	108	2.46 ± 0.98	0.722
Fruits (s/d)	201	1.42 ± 0.93	93	1.46 ± 0.95	108	1.39 ± 0.91	0.689
Milk and dairy products (s/d)	201	2.61 ± 0.80	93	2.59 ± 0.76	108	2.63 ± 0.84	0.850
Meats, fish and eggs (s/d)	201	2.45 ± 0.90	93	2.46 ± 0.91	108	2.44 ± 0.91	0.881
KIDMED index	200	6.50 ± 1.91	93	6.51 ± 1.93	107	6.50 ± 1.90	0.863

TEE, Total Energy Expended; TEI, energy intake; CHD, carbohydrate; PFA, Polyunsaturated Fatty Acids; MFA, Monoinsaturated Fatty Acid; SFA, Saturated Fatty Acid; IPAC, Individual Physical Activity Coefficient; TAWH, Total Active Weekly Hours; s/d: servings/day. Data expressed as mean (x) ± standard deviation (SD).

**Table 2 ijerph-20-00866-t002:** Main diet characteristics of the schoolchildren in accordance with nutritional status (IOTF criteria).

	Normal Weight (*n* = 154)	Overweight (*n* = 36)	Obesity (*n* = 20)	*p*
TEI (kJ/day)	7783.26 ± 1394.20	7816.76 ± 1339.78	7268.28 ± 1632.85	0.308
CHD (%TEI)	44.42 ± 5.41	44.63 ± 5.01	44.58 ± 4.92	0.997
Simple sugar (%TEI)	19.91 ± 3.97	20.63 ± 4.69	21.39 ± 3.89	0.310
Fiber (%TEI)	18.14 ± 5.59	18.74 ± 7.35	16.96 ± 4.67	0.731
Fats (%TEI)	38.90 ± 5.06	39.44 ± 4.94	38.55 ± 4.86	0.815
PFA (%TEI)	4.75 ± 1.43	5.20 ± 1.67	5.02 ± 1.75	0.286
MFA (%TEI)	17.18 ± 3.27	17.42 ± 3.14	17.26 ± 3.29	0.928
SFA (%TEI)	13.37 ± 2.22	13.20 ± 2.56	12.77 ± 2.20	0.598
Proteins (%TEI)	16.67 ± 2.20	15.92 ± 2.08	16.86 ± 2.08	0.193
Cereals and grains (s/d)	4.12 ± 1.47	4.02 ± 1.50	3.45 ± 0.98	0.249
Vegetables (s/d)	2.45 ± 1.05	2.46 ± 1.22	2.65 ± 0.92	0.768
Fruits (s/d)	1.38 ± 0.92	1.49 ± 0.90	1.74 ± 1.01	0.331
Milk and dairy products (s/d)	2.70 ± 0.80	2.30 ± 0.76	2.29 ± 0.67	0.011 ^†^
Meat and meat products (s/d)	2.42 ± 0.89	2.47 ± 0.83	2.75 ± 1.11	0.697

TEI, total energy intake; CHD, carbohydrate; PFA, polyunsaturated fatty acids; MFA, monoinsaturated fatty acid; SFA, saturated fatty acid; s/d, servings/day. Data expressed as mean ± standard deviation. † pos-hoc: Normal weight vs Overweight: *p* = 0.031; Normal weight vs Obesity: *p* = 0.140; Normal weight vs Obesity: *p* = 0.999.

**Table 3 ijerph-20-00866-t003:** Physical activity and sleep length of the schoolchildren in accordance with nutritional status.

	Normal Weight (*n* = 156)	Overweight (*n* = 31)	Obesity (*n* = 15)	*p*
IPAC	1.58 ± 0.11	1.57 ± 0.08	1.62 ± 0.15	0.489
Sleep length (hours)	9.88 ± 1.06	10.29 ± 1.20	9.61 ± 1.00	0.050
TAWH (hours)	3.75 ± 1. 73	3.64 ± 2.06	4.05 ± 2.07	0.601

IPAC, Individual Physical Activity Coefficient; TAWH, Total Active Weekly Hours. Data expressed as mean ± standard deviation.

## Data Availability

Not applicable.

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
