# Peer review of "Dietary and Physical Activity Habits as Conditioning Factors of Nutritional Status among Children of GENYAL Study"

_ijerph, 2023, doi:10.3390/ijerph20010866_

Round 1

Reviewer 1 Report

The study gives an overview of nutritional status (assessed by BMI) of a small sample of 221 school children 6-7 years old in Spain cities, and its association of with various nutritional and lifestyle determinants: total energy intake and energy disbalance, macronutrient and sugar intakes, intakes of certain food groups, number of meals, quality of breakfast, Mediterranean dietary pattern adherence, duration of breastfeeding in the past, physical activity level, sleep duration, and time spent in front of TV and other electronic devices, as well as availability of TV and other electronic devices in the child's room. Even though it was not presented in the best way, seems that study showed that reported nutrient intakes were in an inverse association with BMI, as well as the number of dairy products servings per day, Mediterranean dietary pattern adherence and time of sleep (? – borderline significance) were in an inverse association with BMI, while availability of TV and other electronic devices in the child's room were in a direct association with the children BMI.

The main shortcomings are a small sample size, no sample size calculation, and I am not sure about calculation of certain parameters and about statistics, which tests were where performed, particularly for linear and logistic regression, what was the method (enter or stepwise), what was a dependent variable and what was independent variable, and what were covariates in each linear/logistic regression. The Bonferroni correction was applied, but actuality, the number of covariates (predictors) in each model was not specified, so it is not clear how the corrected p value was calculated, and what was the corrected p value for each regression analysis. All linear/logistic regression models should be given as tables, with all necessary data and parameters included.

Additionally, Figure 1 is missing in the manuscript.

In general, more specific terms should be used and much better and detailed description of methods (please, see below).

In details:

Abstract: also the age of the subjects should be specified, not only the school grade (in different countries a different age of students can be in 1st and 2nd school grades)  

 Line 30: “energy balance” should be defined in the abstract as “ratio of estimated energy intake to estimated energy expenditure (%)” (or “ratio of estimated energy intake to estimated energy requirements (%)”). I would also suggest to use %EB instead of EB in the whole document, since it is more clear that it is a percentage of actual energy intake against energy requirements/expenditure. Also, it should be more specified, if %EB represents actually % of energy disbalance (%ED), or energy overconsumption (in case logistic regression was performed). Therefore, I recommend using term “% of energy disbalance” instead of “energy balance”.

Line 31: Pleas, define the abbreviation if KIDMED score in the abstract and to what it refer. (The “KIDMED Mediterranean Diet Quality Index” was utilized for the purpose of evaluating their adhesion to the Mediterranean eating pattern) – which questions were included in this score should be specified in the methods.

Line 33: “A protective effect against EW depending on the number of dairy servings/day (OR=0.48 (0.29-0.75, p adjusted =0.05) was identified”, is quite unclear and should be “The higher number of dairy servings/day had a protective effect against energy overconsumption (OR=0.48 (0.29-0.75, p adjusted =0.05)”, or “The number of dairy servings/day was in a negative association with %ED  (β=0.48 (0.29-0.75, p adjusted =0.05)” – was logistic or linear regression used here? If logistic (since OR was given), with dichotomous outcomes, then it should be defined in methods what was a dependent variable (“energy overconsumption”?, how it was defied?)

Introduction:

Line 50: “and premature adult mortality” should be “which are associated with premature adult mortality” (mortality is not a disease)

Line 53: “genetic susceptibility plays a key role, and it is… ” should be “genetic susceptibility plays a certain role, but it is …” (studies show that there is not a “key” role of genetic in obesity development, much more environmental factors play role, including epigenetic.. )

Methods:

Line 107: “a 48-hour food record” should be “a twice repeated 24-hour food record” (since 2 separated 24h records were made on 2 non-consecutive days)

Line 113: “TEE was estimated from the WHO equation [15] and Individual Physical  Activity Coefficient (IPAC)).”

Please, be more detailed/precise and give more information which columns and values from table 23 in reference 15 were used for calculation? Values which included only sex and age, or sex, age and body weight? Please, specify in methods which coefficients were used for girls and boys.

Line 121: “A 48-hour physical activity questionnaire” should be “A twice repeated 24-hour physical activity questionnaire” (see above)

Lines 116-119: Please, give more details on “KIDMED Mediterranean Diet Quality Index”-questionnaire and which questions were included (describe in short to what those 16 questions included were referring at). What were the cut-offs for good-quality, low-quality and very-low-quality diet?

Why Bonferroni correction was used for correcting linear regressions? There are pros and cons for its usage (decrease in type 1 error, but increase in type 2 error), which at least should be discussed in study limitations.  

Which method was used for regression analyses, enter of stepwise?

Was the number of subjects in all nutritional categories subgroups adequate enough to perform adequate statistical tests and draw relevant conclusions?

Why study sample calculation was not performed? There is no an adequate explanation. At least, the post hoc analyses of study power should be performed.

Results:

Were there underweight kids in the cohort?

Line 159: “considering the age energy requirements” – were their energy requirements calculated only according to age, or also according to sex, anthropometrics (weight?) and physical activity level?

“TEE was estimated from the WHO equation [15] and Individual Physical  Activity Coefficient (IPAC)).”

Please, be more detailed/precise and give more information both here and in the methods (line 113).

Lines 160-161: “When the relationship between the EB and schoolchildren´s nutritional status was evaluated, it was found an inverse and significant association (β= -1.49 161 (-1.9--1.07), p= 1.53E+05 (p<0.01). “ It is unclear which regression analysis was performed, with “nutritional status categories” or with BMI included? What was a predictor, what was a dependent variable? Please, specify in the sentence.

Lines 174-176:    I do not see the figure 1. Is it missing?

(Figure 1. Amount of servings/day for the different food groups according to the schoolchildren´s  nutritional status. (* p=0.039 Difference between the number of milk and dairy products servings/day according to nutritional status. Data adjusted by age and sex).

Line 183: Define “Beikost” (“an infant's first non-liquid food”)

Line 190-191: “According to Individual Physical Activity Coefficient (IPAC), more than half of the children were classified as sedentary or low-active subjects.” – give exact percentage.

Line 191: “Also, the coefficient was lower during weekends.”- give numbers (mean or median) and p values

Lines 195-200: Was mobile telephone usage for games covered also by this section?

Discussion:

Line 207: “The child’s caloric intake obtained from the dietary questionnaires, showed a paradoxical inverse association between total energy intake and BMI,” – I haven’t seen it in the results section, only the association of “%ED” with “nutritional status”. No association of total energy intake with BMI was shown.

Line 214: “In this work, the diet´s distribution of macronutrients and the simple sugar contribution were far from the recommendation [25].” – which macronutrient’s intake was not adequate? Why this was not emphasized in the Results?

217: What is meant by “nutritional status”: BMI or “nutritional status categories”? Please, specify. In children, simple BMI is not an adequate measure for nutritional status.

Study limitations should be more explored. E.g., no study power or sample size calculation, Bonferroni correction can increase the risk for type 2 error, no other confounding factors included (e.g., information on the presence of overweight/obesity/underweight in the family).

Author Response

Reviewer 1

Comments and Suggestions for Authors

The study gives an overview of nutritional status (assessed by BMI) of a small sample of 221 school children 6-7 years old in Spain cities, and its association of with various nutritional and lifestyle determinants: total energy intake and energy disbalance, macronutrient and sugar intakes, intakes of certain food groups, number of meals, quality of breakfast, Mediterranean dietary pattern adherence, duration of breastfeeding in the past, physical activity level, sleep duration, and time spent in front of TV and other electronic devices, as well as availability of TV and other electronic devices in the child's room. Even though it was not presented in the best way, seems that study showed that reported nutrient intakes were in an inverse association with BMI, as well as the number of dairy products servings per day, Mediterranean dietary pattern adherence and time of sleep (? – borderline significance) were in an inverse association with BMI, while availability of TV and other electronic devices in the child's room were in a direct association with the children BMI.

The main shortcomings are a small sample size, no sample size calculation, and I am not sure about calculation of certain parameters and about statistics, which tests were where performed, particularly for linear and logistic regression, what was the method (enter or stepwise), what was a dependent variable and what was independent variable, and what were covariates in each linear/logistic regression. The Bonferroni correction was applied, but actuality, the number of covariates (predictors) in each model was not specified, so it is not clear how the corrected p value was calculated, and what was the corrected p value for each regression analysis. All linear/logistic regression models should be given as tables, with all necessary data and parameters included.

The authors reflect the small sample size as a weakness of their study in the discussion (Ln 522-24), it may seem small, but it was the maximum we could achieve due to the nature of this study. The authors explain that it is a 5-year longitudinal study, which affects the commitment of the families to participate in the study. According to the referee suggestion, this information has added in the study population and design section (Ln 101-112): “Given the large number of endpoints and associations analyzed, and the absence of initial guesses for the variability for many of them, it was not possible a rigorous and univocal estimation of the sample size. We therefore decided to use the largest sample possible given the available resources”.

The sample size calculation could not be defined due to the characteristics and design of this study. The nature of a clinical trial, the high number of variables of a varied nature and the duration of 5 years made its establishment difficult. The Consejería de Educación e Investigación de la Comunidad de Madrid selected six primary schools in the City of Madrid, two in the north, two in the centre and two in the south. This selection was based on the number of students per school and the socioeconomic level of the neighbourhoods, achieving a representative selection of the average income of the city households. According to the referee's suggestion, this information has been added or improved in the manuscript:

- (Ln 101-112): “Given the large number of endpoints and associations analyzed, and the absence of initial guesses for the variability for many of them, it was not possible a rigorous and univocal estimation of the sample size. We therefore decided to use the largest sample possible given the available resources”.

- (Ln 117-24): “The Consejería de Educación e Investigación de la Comunidad de Madrid was responsible for the selection of six primary schools in the City of Madrid (Spain), two in the north, two in the centre and two in the south. This selection was based on the number of students per school and the socioeconomic level of the neighbourhoods, achieving a representative selection of the average income of the city households. The sample size calculation could not be defined due to the characteristics and design of this study, where we had no previous data for the variability of the outcomes in the population, this being a study of exploratory character

- (Ln 524-35): “On the other hand, we consider that the lack of sample size calculation is one of the weak aspects in our study. Due to the nature of the study as a clinical trial, the large number of variables used, and the duration of 5 years, a statistically robust sample size could not be implemented. Furthermore, the “Consejería de Educación e Investigación de la Co-munidad de Madrid” was responsible for the selection of 6 representative schools of the Community of Madrid (CM) (Spain), considering the number of students per center and the average socioeconomic level of the districts and neighborhoods. Therefore, the se-lection was representative of the average income of the CM households. The lack of so-ciodemographic variables of the characteristics of the families is considered a limitation of this study. Nevertheless, in the GENYAL study, these variables were collected and will be evaluated within other studies with framed objectives on this subject.”

We did not use variable selection approaches in the regression models (this could be considered the “enter” method). This is explained in Materials & Methods and this information has been improved in the new version of the attached manuscript according to the referee suggestions (Ln 223-29) “The associations between anthropometric, physical activity and nutritional data were established through logistic and linear regressions/ANOVA adjusted by sex and age. No variable selection approaches were used (e.g. stepwise), and for each association tested a new regression model was developed with the two involved variables included (one as predictor and the other as outcome) plus the two adjustment variables as additional predictors”.

(Ln 227-30): “p.values were corrected by means of the Bonferroni method. The use of this correction was motivated by computational simplicity and the small number of test performed. Overweight and obesity categories were unified in some analyses as a single category called excess weight (EW)”.

(Ln 540-5):“From a statistical point of view, the use of a simple conservative Bonferroni correction, that controls the family-wise error rate, can lead to an increased rate of false negatives com-pared to other methods, but given the small number of test performed, this should not be a significant issue, and in addition we are more confident in the positive test obtained, without the need of additional confirmatory experiments as those required with e.g. ap-proaches to control the false discovery rate”.

Additionally, Figure 1 is missing in the manuscript.

We appreciate the reviewer's comment. Figure 1 has been relocated closer to its naming in the body of the text.

In general, more specific terms should be used and much better and detailed description of methods (please, see below).

In details:

Abstract: also the age of the subjects should be specified, not only the school grade (in different countries a different age of students can be in 1st and 2nd school grades)  

According to the referee suggestion, the age of the children has been added in the abstract: 6–8 years old (Ln 26).

 Line 30: “energy balance” should be defined in the abstract as “ratio of estimated energy intake to estimated energy expenditure (%)” (or “ratio of estimated energy intake to estimated energy requirements (%)”). I would also suggest to use %EB instead of EB in the whole document, since it is more clear that it is a percentage of actual energy intake against energy requirements/expenditure. Also, it should be more specified, if %EB represents actually % of energy disbalance (%ED), or energy overconsumption (in case logistic regression was performed). Therefore, I recommend using term “% of energy disbalance” instead of “energy balance”.

According to the referee suggestion, the definition of energy has been added in the abstract: energy balance, defined as the radio of estimated energy intake to estimated energy expenditure (%) (Ln 32-3). Likewise, %EB was used instead of EB in the whole document.

Line 31: Pleas, define the abbreviation if KIDMED score in the abstract and to what it refer. (The “KIDMED Mediterranean Diet Quality Index” was utilized for the purpose of evaluating their adhesion to the Mediterranean eating pattern) – which questions were included in this score should be specified in the methods.

Regarding the referee suggestion, the meaning of KIDMED was added in the abstract (Ln 33). In the same way the questions and the scoring of each one was added in the methods section (Ln 182-97): “This questionnaire included 16 dichotomous questions with an affirmative or negative answer, obtaining a final score which ranged from 0 to 12. The questions included were: (Q1) Takes a fruit or fruit juice every day; (Q2) Has a second fruit every day; (Q3) Has fresh or cooked vegetables regularly once a day; (Q4) Has fresh or cooked vegetables more than once a day; (Q5) Consumes fish regularly (at least 2–3 times per week); (Q6) Goes more than once a week to a fast-food (hamburger) restaurant; (Q7) Likes pulses and eats them more than once a week; (Q8) Consumes pasta or rice almost every day (5 or more times per week); (Q9) Has cereals or grains (bread, etc.) for breakfast; (Q10) Consumes nuts regularly (at least 2–3 times per week); (Q11) Uses olive oil at home; (Q12) Skips breakfast; (Q13) Has a dairy product for breakfast (yoghurt, milk, etc.); (Q14) Has commercially baked goods or pastries for breakfast; (Q15) Takes two yoghurts and/or some cheese (40 g) daily; (Q16) Takes sweets and candy several times every day. Questions denoting a negative connotation with respect to the Mediterranean diet were assigned a value of -1 (questions Q6, Q12 and Q16), and those with a positive aspect +1. The sums of the values from the administered test were classified into three levels: (1) >8, optimal Mediterranean diet; (2) 4–7, improvement needed to adjust intake to Mediterranean patterns; (3) ≤3, very low diet quality [25]”.

Line 33: “A protective effect against EW depending on the number of dairy servings/day (OR=0.48 (0.29-0.75, p adjusted =0.05) was identified”, is quite unclear and should be “The higher number of dairy servings/day had a protective effect against energy overconsumption (OR=0.48 (0.29-0.75, p adjusted =0.05)”, or “The number of dairy servings/day was in a negative association with %ED  (β=0.48 (0.29-0.75, p adjusted =0.05)” – was logistic or linear regression used here? If logistic (since OR was given), with dichotomous outcomes, then it should be defined in methods what was a dependent variable (“energy overconsumption”?, how it was defied?)

According to our results, a protective effect against EW=excess weight (not overconsumption or %ED, it was probably a misunderstanding) depending on the number of dairy servings/day (OR=0.48 (0.29-0.75, p adjusted =0.05) was identified.

To obtain this result a logistic regression was performed in which normal or excess weight was the dependent variable and the number of dairy servings/day the independent variable. For each increase in the number of servings, the risk to develop excess weight is reduced by 52% (OR=0.48). When the sample of schoolchildren was divided into normal weight and ponderal overload (overweight + obesity=excess weight), the number of dairy servings per day showed a protective effect against overload (OR=0.48 (0.29-0.75, p = 0.0014) (data adjusted for sex and age). This effect remained significant after adjusting for the 20 nutritional variables analyzed in the study and maintained a trend towards significance after multiple comparison analysis that included a total of 50 variables from different categories included in the study (nutritional, anthropometric, socio-sanitary, etc.) (p = 0.05).

According to the referee suggestion this phrase was introduced in the statistical analyses section (Ln 230-1): “The multiple comparison analysis included a total of 50 variables from different categories included in the study (nutritional, anthropometric, socio-sanitary, etc.)”

Introduction:

Line 50: “and premature adult mortality” should be “which are associated with premature adult mortality” (mortality is not a disease)

According to the referee suggestion this phrase was replaced in the Introduction (Ln 56).

Line 53: “genetic susceptibility plays a key role, and it is… ” should be “genetic susceptibility plays a certain role, but it is …” (studies show that there is not a “key” role of genetic in obesity development, much more environmental factors play role, including epigenetic.. )

According to the referee suggestion this phrase was replaced in the Introduction (Ln 60).

Methods:

Line 107: “a 48-hour food record” should be “a twice repeated 24-hour food record” (since 2 separated 24h records were made on 2 non-consecutive days)

According to the referee suggestion this phrase was replaced in the Introduction (Ln 150).

Line 113: “TEE was estimated from the WHO equation [15] and Individual Physical  Activity Coefficient (IPAC)).”

Please, be more detailed/precise and give more information which columns and values from table 23 in reference 15 were used for calculation? Values which included only sex and age, or sex, age and body weight? Please, specify in methods which coefficients were used for girls and boys.

According to the referee suggestion this information has been improved in the new version of the attached manuscript: “The energy balance (EB) was calculated as total energy intake (TEI) (kcal/day) / total energy expended (TEE) (kcal/day) x 100 (where TEI was obtained from the DIAL software and TEE was estimated from the WHO equation [15] that establish a basal expenditure based on the weight, age and sex of each individual and then, this basal expenditure, is multiplied by a Individual Physical Activity Coefficient (IPAC))” (Ln:154-8).

Line 121: “A 48-hour physical activity questionnaire” should be “A twice repeated 24-hour physical activity questionnaire” (see above)

According to the referee suggestion this phrase was replaced in the Introduction (Ln 199).

Lines 116-119: Please, give more details on “KIDMED Mediterranean Diet Quality Index”-questionnaire and which questions were included (describe in short to what those 16 questions included were referring at). What were the cut-offs for good-quality, low-quality and very-low-quality diet?

To clarify this aspect in the manuscript and avoid going to the source article, the following sentences have been added (Ln 182-97): “This questionnaire included 16 dichotomous questions with an affirmative or negative answer, obtaining a final score which ranged from 0 to 12. The questions included were: (Q1) Takes a fruit or fruit juice every day; (Q2) Has a second fruit every day; (Q3) Has fresh or cooked vegetables regularly once a day; (Q4) Has fresh or cooked vegetables more than once a day; (Q5) Consumes fish regularly (at least 2–3 times per week); (Q6) Goes more than once a week to a fast-food (hamburger) restaurant; (Q7) Likes pulses and eats them more than once a week; (Q8) Consumes pasta or rice almost every day (5 or more times per week); (Q9) Has cereals or grains (bread, etc.) for breakfast; (Q10) Consumes nuts regularly (at least 2–3 times per week); (Q11) Uses olive oil at home; (Q12) Skips breakfast; (Q13) Has a dairy product for breakfast (yoghurt, milk, etc.); (Q14) Has commercially baked goods or pastries for breakfast; (Q15) Takes two yoghurts and/or some cheese (40 g) daily; (Q16) Takes sweets and candy several times every day. Questions denoting a negative connotation with respect to the Mediterranean diet were assigned a value of -1 (questions Q6, Q12 and Q16), and those with a positive aspect +1. The sums of the values from the administered test were classified into three levels: (1) >8, optimal Mediterranean diet; (2) 4–7, improvement needed to adjust intake to Mediterranean patterns; (3) ≤3, very low diet quality [25].”

Why Bonferroni correction was used for correcting linear regressions? There are pros and cons for its usage (decrease in type 1 error, but increase in type 2 error), which at least should be discussed in study limitations.  

We basically used Bonferroni correction because of its little computational complexity and the small number of test that resulted in a negligible issue in terms of false negatives. This has been better explained in the text:

  • In Materials & Methods (Ln 228-30): “pvalues were corrected by means of the Bonferroni method. The use of this correction was motivated by computational simplicity and the small number of test performed.”
  • In Discussion/limitations (Ln 542-7): “From a statistical point of view, the use of a simple conservative Bonferroni correction, that controls the family-wise error rate, can lead to an increased rate of false negatives com-pared to other methods, but given the small number of test performed, this should not be a significant issue, and in addition we are more confident in the positive test obtained, without the need of additional confirmatory experiments as those required with e.g. ap-proaches to control the false discovery rate”.

 Which method was used for regression analyses, enter of stepwise?

We did not use variable selection approaches in the regression models (this could be considered the “enter” method). To clarify this aspect in the manuscript, the following sentences have been added in Materials & Methods (ln 223-8): “The associations between anthropometric, physical activity and nutritional data were established through logistic and linear regressions/ANOVA adjusted by sex and age. No variable selection approaches were used (e.g. stepwise), and for each association tested a new regression model was developed with the two involved variables included (one as predictor and the other as outcome) plus the two adjustment variables as additional predictors”.

Was the number of subjects in all nutritional categories subgroups adequate enough to perform adequate statistical tests and draw relevant conclusions?

The main dietary characteristics of schoolchildren according to their nutritional status were summarized in Table 2 mainly to have a broader description of the sample. Given the fact that the number of children with obesity was reduced, the results of overweight and obesity rates were unified as a single category called excess weight (EW).

The authors are aware of the limitation of the sample size of the study, and to make it better reflected, the following text has been added after the conclusions (Ln 562-3): “However, the small sample size of this study requires that its results be corroborated in future studies with larger sample sizes”.

Why study sample calculation was not performed? There is no an adequate explanation. At least, the post hoc analyses of study power should be performed.

This work forms part of GENYAL study to childhood obesity prevention. The main objective of this study is to design and validate a predictive model that identifies those children who would benefit most from actions aimed at reducing the risk of obesity and its complications, considering both environmental and genetic factors. Given the large number of endpoints and associations analyzed, and the absence of initial guesses for the variability for many of them, it was not possible a rigorous and univocal estimation of the sample size. We therefore decided to use the largest sample possible given the available resources. The results presented in this manuscript were a cross-sectional evaluation carried out in the context of the global project.

To clarify this aspect in the manuscript the following sentence has been introduced in the Study population and design section (Ln 102-13):“Given the large number of endpoints and associations analyzed, and the absence of initial guesses for the variability for many of them, it was not possible a rigorous and univocal estimation of the sample size. We therefore decided to use the largest sample possible given the available resources”

Results:

Were there underweight kids in the cohort?

Yes, 5% of children with underweight were found in the sample following the IOTF criteria. Given the small sample size, they could not be treated independently.

Line 159: “considering the age energy requirements” – were their energy requirements calculated only according to age, or also according to sex, anthropometrics (weight?) and physical activity level?

According to the referee suggestion this phrase was replaced (Ln 269-71): “Regarding schoolchildren´s energy balance (EB), calculated as TEI (kcal/day) / TEE (kcal/day) x 100, 38.8% of the children were overfeeding while 28.9% were classified as having insufficient intake”.

The authors also modified the methodology to better explain the derivation of these variables (Ln: 154-8): The energy balance (%EB) was calculated as total energy intake (TEI) (kcal/day) / total energy expended (TEE) (kcal/day) x 100 (where TEI was obtained from the DIAL software and TEE was estimated from the WHO equation [24] that establish a basal expenditure based on the weight, age and sex of each individual and then, this basal expenditure, is multiplied by a Individual Physical Activity Coefficient (IPAC))” (ln:174).

Lines 160-161: “When the relationship between the EB and schoolchildren´s nutritional status was evaluated, it was found an inverse and significant association (β= -1.49 161 (-1.9--1.07), p= 1.53E+05 (p<0.01). “ It is unclear which regression analysis was performed, with “nutritional status categories” or with BMI included? What was a predictor, what was a dependent variable? Please, specify in the sentence.

To obtain this result a lineal regression was performed in which BMI was the dependent variable and the EB was the independent variable. The more the energy intake exceeds the energy expenditure (over-intake evaluated through the EB variable), the BMI is reduced by 1.49 kg/m2

According to the referee suggestion this phrase was changed (ln 271-4): When the relationship between the %EB (independent variable) and schoolchildren´s nutritional status (BMI) (dependent variable) was evaluated, it was found an inverse and significant association (β= -1.49 (-1.9--1.07), p= 1.53E+05 (p<0.01)”.

Lines 174-176:    I do not see the figure 1. Is it missing?

(Figure 1. Amount of servings/day for the different food groups according to the schoolchildren´s  nutritional status. (* p=0.039 Difference between the number of milk and dairy products servings/day according to nutritional status. Data adjusted by age and sex).

We do not really know the problem because the figure was in the article. We hope you do not have problems to see the figure in the new version

Line 183: Define “Beikost” (“an infant's first non-liquid food”)

Regarding the referee suggestion we introduce this definition “defined as an infant's first non-liquid food” (Ln 400).

Line 190-191: “According to Individual Physical Activity Coefficient (IPAC), more than half of the children were classified as sedentary or low-active subjects.” – give exact percentage.

The exact percentage 55,6% was included instead of “more than half” in Ln 429.

Line 191: “Also, the coefficient was lower during weekends.”- give numbers (mean or median) and p values

Regarding the referee suggestion we introduce this information (Ln 431): “(1.52±0.13 in the weekend and 1.61±0.12 during the week, p<0,001)”.

Lines 195-200: Was mobile telephone usage for games covered also by this section?

No, unfortunately this information was not collected

Discussion:

Line 207: “The child’s caloric intake obtained from the dietary questionnaires, showed a paradoxical inverse association between total energy intake and BMI,” – I haven’t seen it in the results section, only the association of “%ED” with “nutritional status”. No association of total energy intake with BMI was shown.

Thanks to the reviewer's suggestion, the authors have changed the name of the variable and the information (Ln: 446): “The child’s energy balance (TEI (kcal/day)/TEE (kcal/day), showed a paradoxical inverse association with the BMI”.

Line 214: “In this work, the diet´s distribution of macronutrients and the simple sugar contribution were far from the recommendation [25].” – which macronutrient’s intake was not adequate? Why this was not emphasized in the Results?

Regarding the referee suggestion we introduce this information in the results (Ln 260-4):The percentages of mean values of nutrient intake over the total caloric value were 44.48% in carbohydrates, 20.15% simple sugars, 18.16% fiber, 16.55% protein, 38.96% of fats,13.28% SFA, 17.22% MFA and 4.84% PFA”, and this information in the discussion (ln:452-58): In this work, the diet´s distribution of proteins and fats and the simple sugar con-tribution were far from the recommendation and they fall short on total carbohydrates [34]. This has been previously observed in different national children surveys [35] and in the EsNuPi study [11]. These data reflect that the diet in children must be improved. Excess protein in the diet have previously been associated with obesity and other met-abolic risks at an early age and higher BMI later in childhood [36] and seems to maintain the same trend in terms of excess fat [37].”

217: What is meant by “nutritional status”: BMI or “nutritional status categories”? Please, specify. In children, simple BMI is not an adequate measure for nutritional status.

With the term "nutritional status" the authors referred to both concepts, BMI and nutritional status categories. To specify the information, the following changes have been made in the text (Ln 458-60): “Nevertheless, these aspects of the diet were not associated with the schoolchildren´s nutritional status categories or with the BMI

Study limitations should be more explored. E.g., no study power or sample size calculation, Bonferroni correction can increase the risk for type 2 error, no other confounding factors included (e.g., information on the presence of overweight/obesity/underweight in the family).

Regarding the referee suggestion the authors have improved the information regarding the limitations section (Ln: 530-8): “On the other hand, we consider that the lack of sample size calculation is one of the weak aspects in our study. Due to the nature of the study as a clinical trial, the large number of variables used, and the duration of 5 years, a statistically robust sample size could not be implemented. Furthermore, the “Consejería de Educación e Investigación de la Comunidad de Madrid” was responsible for the selection of 6 representative schools of the Community of Madrid (CM) (Spain), considering the number of students per center and the average socioeconomic level of the districts and neighborhoods. Therefore, the selection was representative of the average income of the CM households.

And  (Ln 542-7):”From a statistical point of view, the use of a simple conservative Bonferroni correction, that controls the family-wise error rate, can lead to an increased rate of false negatives compared to other methods, but given the small number of test performed, this should not be a significant issue, and in addition we are more confident in the positive test obtained, without the need of additional confirmatory experiments as those required with e.g. approaches to control the false discovery rate.”

Reviewer 2 Report

The study is current and relevant, but needs extensive review to be considered for publication, as per the following comments:

Introduction

The introduction could be improved with the inclusion of more recent citations, of epidemiological studies on the global and Spanish prevalence of childhood obesity and physical inactivity, as well as their relationship with the early onset of chronic non-communicable diseases.

Methods

Regarding the calculation of the sample size, regardless of the difficulties, it could have been carried out a priori, but as it was not carried out, the authors must present the power of the sample calculated a posteriori.

It is necessary to further detail the selection of schools:

1. What are the criteria for selecting schools

2. How was the students/school’s stratification

3. Is the sample representative of the school population in the age group of interest in the study?

The statistical test used to verify the assumption of normal data distribution should be mentioned.

Results

The presentation of the results is poor but could be improved with the presentation of the relationships between nutritional status, food consumption and body composition obtained by bioimpedance. In addition, the relationship between physical activity level and body composition could also be presented.

Why were the results of overweight and obesity initially presented by three different criteria (WHO, IOTF and Orbegozo Foundation), but later the authors opted for the IOTF? What justifies this option?

I did not find Figure 1 mentioned on page 5.

Discussion

The discussion should be improved with the inclusion of more recent studies that corroborate or not the results found in the study.

Author Response

The study is current and relevant, but needs extensive review to be considered for publication, as per the following comments:

Introduction

The introduction could be improved with the inclusion of more recent citations, of epidemiological studies on the global and Spanish prevalence of childhood obesity and physical inactivity, as well as their relationship with the early onset of chronic non-communicable diseases.

Regarding the referee´s suggestion the authors have improved the information regarding the introduction section (Ln 74-91) and added new references: “In Spain, very few studies have been carried out on eating habits in the pediatric population to fight against childhood obesity. Almost all of them are descriptive and cross-sectional studies. The ALADINO study [4] was carried out with a representative sample of the school population from 6 to 9 years of age, and the EsNuPi study [11] was carried out with a representative sample of Spanish children from 1 to 10 years of age residing in urban areas (except Ceuta and Melilla). There are the two most complete national studies on childhood age about their specific objectives related to the study of the prevalence of obesity and the study associated environmental factors. However, these studies do not include nutritional intervention. The THAO Child Health Program [12] and the POIBC study [13] were studies that collected data on nutritional status and lifestyle, with community nutritional intervention in Spain, but not carried out in the social context of the city of Madrid.

Several studies have been conducted in other countries to study the association between nutritional and physical activity factors and excess weight in schoolchildren, underlining the need for more research in this area [14, 15, 16, 17].

The study of environmental factors concerning nutritional status is of great interest since they have indisputable importance in the pathogenesis of obesity and have been responsible for being able to define a classical Western society as "obesogenic" [18].”

Methods

Regarding the calculation of the sample size, regardless of the difficulties, it could have been carried out a priori, but as it was not carried out, the authors must present the power of the sample calculated a posteriori.

The calculation of power after test have been performed (retrospective, post hoc or observed power) has been shown to be logically invalid, contains no new information (as there is a one-to-one relationship with the obtained pvalues) and can lead to misinterpretations of the results. See for instance:

(1)          Hoenig, J. M.; Heisey, D. M. The Abuse of Power: The Pervasive Fallacy of Power Calculations for Data Analysis. The American Statistician 200155 (1), 19–24. https://doi.org/10.1198/000313001300339897.

(2)          Dziak, J. J.; Dierker, L. C.; Abar, B. The Interpretation of Statistical Power after the Data Have Been Gathered. Curr Psychol 2018. https://doi.org/10.1007/s12144-018-0018-1.

It is necessary to further detail the selection of schools:

  1. What are the criteria for selecting schools

The “Consejería de Educación e Investigación de la Comunidad de Madrid” was responsible for the selection of 6 representative schools of the Community of Madrid (CM) (Spain). According to the referee suggestion this phrase has been improved (Ln 118-22): “The Consejería de Educación e Investigación de la Comunidad de Madrid was responsible for the selection of six primary schools in the City of Madrid (Spain), two in the north, two in the centre and two in the south. This selection was based on the number of students per school and the socioeconomic level of the neighbourhoods, achieving a representative selection of the average income of the city households”.

  1. How was the students/school’s stratification

To clarify this aspect in the manuscript the following sentence has been introduced in the Study population and design section (Ln 130-4): “For the nutritional intervention, randomization was carried out by school center. Thus, participating schools were randomly and proportionally stratified into 2 groups: intervention schools and control schools, considering the number of participants per center, their geographic area and their socioeconomic status. The randomization procedure was carried out with the statistical software R version 3.4 (www.r-project.org).”

  1. Is the sample representative of the school population in the age group of interest in the study?

No, the sample size is not representative. The authors explain that it is a 5-year longitudinal study, which affects the commitment of the families to participate in the study. The sample size calculation could not be defined due to the characteristics and design of this study. The nature of a clinical trial, the high number of variables of a varied nature and the duration of 5 years made its establishment difficult. The Consejería de Educación e Investigación de la Comunidad de Madrid selected six primary schools in the City of Madrid, two in the north, two in the centre and two in the south. This selection was based on the number of students per school and the socioeconomic level of the neighbourhoods, achieving a representative selection of the average income of the city households (Ln 118-22). The authors reflect the small sample size as a weakness of their study in the discussion (Ln 528). It may seem small, but it was the maximum we could achieve due to the nature of this study. Also, the following text has been added to the study population and design section (Ln 122-25): “The sample size calculation could not be defined due to the characteristics and design of this study, where we had no previous data for the variability of the outcomes in the population, this being a study of exploratory character”.

The statistical test used to verify the assumption of normal data distribution should be mentioned.

Regarding the referee suggestion we introduce this information on the statistical analyses point (Ln 219): “Shapiro-Wilk test (p > 0.05) was used to assess the normality of data”.

Results

The presentation of the results is poor but could be improved with the presentation of the relationships between nutritional status, food consumption and body composition obtained by bioimpedance. In addition, the relationship between physical activity level and body composition could also be presented.

Regarding the referee suggestion we introduce this information in the results:

- (Ln 271-4): “When the relationship between the %EB (independent variable) and schoolchildren´s BMI and FM (dependent variables) was evaluated, it was found an inverse and significant association (β= -1.49 (-1.9--1.07), p= 1.53E+05 (p<0.01) and β= -3.46 (-4.62--2.29), p= 1.24E+08 (p<0.01), respectively).”.

- Ln 396-8: “The total score obtained in the KIDMED questionnaire was inversely associated with the schoolchildren’s BMI (β= -0.19 (95% IC -0.38-0), p = 0.04) and FM (β= -0.65 (-1.15--0.14), p=0,01).”

Why were the results of overweight and obesity initially presented by three different criteria (WHO, IOTF and Orbegozo Foundation), but later the authors opted for the IOTF? What justifies this option?

The authors wanted to include data according to all three criteria because there are currently no gold standard criteria for defining excess weight. Thus, the data can be easily compared with those of other studies. For the analysis, we opted for the IOTF criteria in the first place because they are validated in the European child population, because they are widely used in other studies and because, of the three options, these were the average results (the results of excess weight according to criteria WHO were the lowest and according to Orbegozo the highest).

I did not find Figure 1 mentioned on page 5.

We do not really know the problem because the figure was in the article. We hope you do not have problems to see the figure in the new version

Discussion

The discussion should be improved with the inclusion of more recent studies that corroborate or not the results found in the study.

Regarding the referee suggestion the authors have improved the information regarding the discusion section: Ln 454-59: “were far from the recommendation and they fall short on total carbohydrates [34]. This has been previously observed in different national children surveys [35] and in the EsNuPi study [11]. These data reflect that the diet in children must be improved. Excess protein in the diet have previously been associated with obesity and other metabolic risks at an early age and higher BMI later in childhood [36] and seems to maintain the same trend in terms of excess fat [37].”, Ln 462-67: “Significant differences were observed between the hours dedicated to practising physical activity weekly (TAWH) between boys and girls, observing that the boys spent the most time. These results have been observed in other studies [14, 17]. In the ALA-DINO 2019 study, this same trend was observed. Besides, boys carried out more extra-curricular sports activities than girls, reflecting an inequitable distribution between genders in sports since childhood [4]”, Ln 471-3 “According to the latest reviews on the subject, milk and dairy products are not consistently associated or are inversely associated, with obesity and indicators of adiposity in children [40].”, Ln 501-5 “and is similar to the results reported by the ALADINO 2019 study [4]. This fact is con-sidered one of the reasons why Spain is among the European countries with the highest prevalence of childhood obesity [44]. However, in the EsNupi study [11] a main dietary pattern was observed related to the Mediterranean diet, with a high consumption of cereals, fruits, vegetables, milk and dairy products.”, Ln 507-10: “In this study we observed that the 22.0% of children spent more than two hours of TV time a day (television, playing video games, computer...), mainly during the weekend. In the ALADINO 2019 study, the percentage was higher, and the same trend was ob-served towards greater inactivity and passive leisure on the screen during weekends [4]”, Ln 515-6: “Moreover, in other studies, total screen time has been directly related to excess weight in children [14].”, Ln 521-25: “It was observed that children with obesity performed more physical activity through the calculated variables (IPAC and TAWH) than children with normal weight and over-weight. The results did not obtain statistical significance. In the ALADINO 2019 study, contrary data was obtained. Children with normal weight were the most active [4].”.

Reviewer 3 Report

The authors (AA) aim to describe the main lifestyle characteristics (diet and exercise) and their possible association with nutritional status in a group of schoolchildren enrolled in the GENYAL study for the prevention of childhood obesity. This is an engaging article with a study design appropriate and useful to increase our knowledge of the issue. The title reports the key features of the paper encouraging the reader to read more.

Addressing the following issues can make this interesting manuscript eligible for the publication.

Abstract

Lines 26-27: AA should report the methods for collecting these data.

Lines 28-29: Specify the different criteria used.

Introduction

The references are relevant and appropriate, but AA should add other references about their topic regard children (6-10 years) in the similar setting also in other countries as reported below:

              Zhu, X.; Haegele, J.A.; Tang, Y.; Wu, X. Prevalence and Demographic Correlates of Overweight, Physical Activity, and Screen Time Among School-Aged Children in Urban China: The Shanghai Study. Asia-Pacific journal of public health 2018, 30, 353 118-127, doi:10.1177/1010539518754538.

              Paduano S, Borsari L, Salvia C, Arletti S, Tripodi A, Pinca J, Borella P. Risk Factors for Overweight and Obesity in Children Attending the First Year of Primary Schools in Modena, Italy. J Community Health. 2020 Apr;45(2):301-309. doi: 10.1007/s10900-019-00741-7.

              Kumar, S.; Kelly, A.S. Review of Childhood Obesity: From Epidemiology, Etiology, and Comorbidities to Clinical Assessment and Treatment. Mayo Clinic proceedings 2017, 92, 251-265, doi:10.1016/j.mayocp.2016.09.017.

              Paduano S., Greco A., Borsari L., Salvia C., Tancredi S., Pinca J., Midili S., Tripodi A., Borella P., Marchesi I. Physical and Sedentary Activities and Childhood Overweight/Obesity: A Cross-Sectional Study among First-Year Children of Primary Schools in Modena, Italy. Int J Environ Res Public Health. 2021 Mar; 18(6): 3221. doi: 10.3390/ijerph18063221.

Materials and Methods

Line 105: AA should update reference about IOTF.

Lines 103-105: Explain the reason for including three different classifications.

Paragraphs 2.3-2.4: AA should better explain the questionnaires or they can attach them.

Why did AA considerer to collect data about socio-demographic characteristics of children family?

Results

AA should report a table with descriptive data of their population.

Lines 181-200: Explain multiple tests.

Lines 185-186: Delete them.

Discussion

Add the absence of socio-demographic characteristics of children family as limitation of the study.

It would be interesting for AA to discuss their findings also by comparing these data with other studies including a similar population in other countries as well.

Author Response

The authors (AA) aim to describe the main lifestyle characteristics (diet and exercise) and their possible association with nutritional status in a group of schoolchildren enrolled in the GENYAL study for the prevention of childhood obesity. This is an engaging article with a study design appropriate and useful to increase our knowledge of the issue. The title reports the key features of the paper encouraging the reader to read more.

Addressing the following issues can make this interesting manuscript eligible for the publication.

Abstract

Lines 26-27: AA should report the methods for collecting these data.

According to the referee suggestion, this information has been added in the abstract (Ln 26-8): “Anthropometric (BMI and bioimpedance), dietary intake (twice repeated 24-hour food record) and physical activity (twice repeated 24-hour physical activity questionnaire) data were collected.”

Lines 28-29: Specify the different criteria used.

According to the referee suggestion, this information has been added in the abstract (Ln: 30-1): “The prevalence of EW was 19%, 25.4% and 32.2% according to Orbegozo Foundation, IOFT and WHO criteria, respectively”.

Introduction

The references are relevant and appropriate, but AA should add other references about their topic regard children (6-10 years) in the similar setting also in other countries as reported below:

⁻              Zhu, X.; Haegele, J.A.; Tang, Y.; Wu, X. Prevalence and Demographic Correlates of Overweight, Physical Activity, and Screen Time Among School-Aged Children in Urban China: The Shanghai Study. Asia-Pacific journal of public health 2018, 30, 353 118-127, doi:10.1177/1010539518754538.

⁻              Paduano S, Borsari L, Salvia C, Arletti S, Tripodi A, Pinca J, Borella P. Risk Factors for Overweight and Obesity in Children Attending the First Year of Primary Schools in Modena, Italy. J Community Health. 2020 Apr;45(2):301-309. doi: 10.1007/s10900-019-00741-7.

⁻              Kumar, S.; Kelly, A.S. Review of Childhood Obesity: From Epidemiology, Etiology, and Comorbidities to Clinical Assessment and Treatment. Mayo Clinic proceedings 2017, 92, 251-265, doi:10.1016/j.mayocp.2016.09.017.

⁻              Paduano S., Greco A., Borsari L., Salvia C., Tancredi S., Pinca J., Midili S., Tripodi A., Borella P., Marchesi I. Physical and Sedentary Activities and Childhood Overweight/Obesity: A Cross-Sectional Study among First-Year Children of Primary Schools in Modena, Italy. Int J Environ Res Public Health. 2021 Mar; 18(6): 3221. doi: 10.3390/ijerph18063221.

Thanks to the reviewer's suggestion, the authors have included other references about their topic regard children (6-10 years) in the similar setting also in other countries (Ln-74-91): “In Spain, very few studies have been carried out on eating habits in the pediatric population to fight against childhood obesity. Almost all of them are descriptive and cross-sectional studies. The ALADINO study [4] was carried out with a representative sample of the school population from 6 to 9 years of age, and the EsNuPi study [11] was carried out with a representative sample of Spanish children from 1 to 10 years of age residing in urban areas (except Ceuta and Melilla). There are the two most complete national studies on childhood age about their specific objectives related to the study of the prevalence of obesity and the study associated environmental factors. However, these studies do not include nutritional intervention. The THAO Child Health Program [12] and the POIBC study [13] were studies that collected data on nutritional status and lifestyle, with community nutritional intervention in Spain, but not carried out in the social context of the city of Madrid.

Several studies have been conducted in other countries to study the association between nutritional and physical activity factors and excess weight in schoolchildren, underlining the need for more research in this area [14, 15, 16, 17].

The study of environmental factors concerning nutritional status is of great interest since they have indisputable importance in the pathogenesis of obesity and have been responsible for being able to define a classical Western society as "obesogenic" [18]”.

Materials and Methods

Line 105: AA should update reference about IOTF.

According to the referee suggestion, the reference has been changed (Ln 627).

Lines 103-105: Explain the reason for including three different classifications.

The authors wanted to include data according to all three criteria because there are currently no gold standard criteria for defining excess weight. Thus, the data can be easily compared with those of other studies. For the analysis, we opted for the IOTF criteria in the first place because they are validated in the European child population, because they are widely used in other studies and because, of the three options, these were the average results (the results of excess weight according to criteria WHO were the lowest and according to Orbegozo the highest).

Paragraphs 2.3-2.4: AA should better explain the questionnaires or they can attach them.

Regarding the referee suggestion, the explanation of the questionnaires have been improved in the manuscript (Ln:182-97): “This questionnaire included 16 dichotomous questions with an affirmative or negative answer, obtaining a final score which ranged from 0 to 12. The questions included were: (Q1) Takes a fruit or fruit juice every day; (Q2) Has a second fruit every day; (Q3) Has fresh or cooked vegetables regularly once a day; (Q4) Has fresh or cooked vegetables more than once a day; (Q5) Consumes fish regularly (at least 2–3 times per week); (Q6) Goes more than once a week to a fast-food (hamburger) restaurant; (Q7) Likes pulses and eats them more than once a week; (Q8) Consumes pasta or rice almost every day (5 or more times per week); (Q9) Has cereals or grains (bread, etc.) for breakfast; (Q10) Consumes nuts regularly (at least 2–3 times per week); (Q11) Uses olive oil at home; (Q12) Skips breakfast; (Q13) Has a dairy product for breakfast (yoghurt, milk, etc.); (Q14) Has commercially baked goods or pastries for breakfast; (Q15) Takes two yoghurts and/or some cheese (40 g) daily; (Q16) Takes sweets and candy several times every day. Questions denoting a negative connotation with respect to the Mediterranean diet were assigned a value of -1 (questions Q6, Q12 and Q16), and those with a positive aspect +1. The sums of the values from the administered test were classified into three levels: (1) >8, optimal Mediterranean diet; (2) 4–7, improvement needed to adjust intake to Mediterranean patterns; (3) ≤3, very low diet quality [16].”

(Ln: 199-203): “A twice repeated 24-hour physical activity questionnaire (a weekday and a weekend day) was collected [17]. The parents had to specify the time that their children spent during 24 hours of a weekday and 24 hours of a weekend day doing different activities, including resting hours and activities with a variable level of intensity (very light, light, moderate and intense).”

Why did AA considerer to collect data about socio-demographic characteristics of children family?

Since the socioeconomic factors of the families are related to the nutritional status of the children (Grassi T et al;2016), the socioeconomic level of the city's neighbourhoods was considered to include a more representative sample.

Grassi, T.; De Donno, A.; Bagordo, F.; Serio, F.; Piscitelli, P.; Ceretti, E.; Zani, C.; Viola, G.C.V.; Villarini, M.; Moretti, M.; Levorato, S.; Carducci, A.; Verani, M.; Donzelli, G.; Bonetta, S.; Bonetta, S.; Carraro, E.; Bonizzoni, S.; Bonetti, A.; Gelatti, U. Socio-Economic and Environmental Factors Associated with Overweight and Obesity in Children Aged 6–8 Years Living in Five Italian Cities (the MAPEC_LIFE Cohort). Int. J. Environ. Res. Public Health 201613, 1002. https://doi.org/10.3390/ijerph13101002

Results

AA should report a table with descriptive data of their population.

Regarding the referee suggestion we have added a table with descriptive data of the anthropometric, physical and leisure activities and intake data of the sample. Table 1 (Ln 252).

Lines 181-200: Explain multiple tests.

Ln 181:“Concerning KIDMED questionnaire, more than 50.0% of the children needed to im-prove their diet and 7.50% had a very low-quality diet. The total score obtained in the KIDMED questionnaire was inversely associated with the schoolchildren’s BMI (β= -0.19 (95% IC -0.38-0), p = 0.04). However, this association lost significance after correction by multiple tests”: This variable was tested within a pool of tests for nutritional, socio-economic and behavior putative predictor variables.

Ln 200: “Concerning relation to this, it was observed a direct and significant association between the availability of these devices and the child’s BMI (β= 1.15 (95% IC 0.20-2.10) p=0.017) and their FM β= 3.28 (95% IC 0,69-5,87) p=0,013. However, these associations lost significance after correction by multiple tests”: In the same pool of tests above described.

Lines 185-186: Delete them.

Regarding the referee suggestion those lines have been correctly included in the body of the text (Ln: 402-4).

Discussion

Add the absence of socio-demographic characteristics of children family as limitation of the study.

Regarding the referee suggestion we introduce this information (Ln 539-42): “The lack of sociodemographic variables of the characteristics of the families is considered a limitation of this study. Nevertheless, in the GENYAL study, these variables were collected and will be evaluated within other studies with framed objectives on this subject”.

It would be interesting for AA to discuss their findings also by comparing these data with other studies including a similar population in other countries as well.

Regarding the referee's suggestion, we have improved the discussion section with other references, including a similar population in other countries and at the national level: Ln 454-59: “were far from the recommendation and they fall short on total carbohydrates [34]. This has been previously observed in different national children surveys [35] and in the EsNuPi study [11]. These data reflect that the diet in children must be improved. Excess protein in the diet have previously been associated with obesity and other metabolic risks at an early age and higher BMI later in childhood [36] and seems to maintain the same trend in terms of excess fat [37].”, Ln 462-67: “Significant differences were observed between the hours dedicated to practising physical activity weekly (TAWH) between boys and girls, observing that the boys spent the most time. These results have been observed in other studies [14, 17]. In the ALA-DINO 2019 study, this same trend was observed. Besides, boys carried out more extra-curricular sports activities than girls, reflecting an inequitable distribution between genders in sports since childhood [4]”, Ln 471-3 “According to the latest reviews on the subject, milk and dairy products are not consistently associated or are inversely associated, with obesity and indicators of adiposity in children [40].”, Ln 501-5 “and is similar to the results reported by the ALADINO 2019 study [4]. This fact is con-sidered one of the reasons why Spain is among the European countries with the highest prevalence of childhood obesity [44]. However, in the EsNupi study [11] a main dietary pattern was observed related to the Mediterranean diet, with a high consumption of cereals, fruits, vegetables, milk and dairy products.”, Ln 507-10: “In this study we observed that the 22.0% of children spent more than two hours of TV time a day (television, playing video games, computer...), mainly during the weekend. In the ALADINO 2019 study, the percentage was higher, and the same trend was ob-served towards greater inactivity and passive leisure on the screen during weekends [4]”, Ln 515-6: “Moreover, in other studies, total screen time has been directly related to excess weight in children [14].”, Ln 521-25: “It was observed that children with obesity performed more physical activity through the calculated variables (IPAC and TAWH) than children with normal weight and over-weight. The results did not obtain statistical significance. In the ALADINO 2019 study, contrary data was obtained. Children with normal weight were the most active [4].”.

Round 2

Reviewer 1 Report

I would like to thank the authors for detailed answers, they were quite satisfactory.

There are just several points which are not clear.

1.       “Were there underweight kids in the cohort?

Yes, 5% of children with underweight were found in the sample following the IOTF criteria. Given the small sample size, they could not be treated independently.”

Were they excluded from further analyses, or were merged with “normal weight”? This information should be given in the text (lines 217-219).

2.       In Table 2, which criteria for definition of obesity/overweight were used? Please, specify in the table.

In the table 2, the numbers of normal weight, overweight and obese (156+31+15=202) do not match with the numbers (percentages) given in the text lines 217-219, nor with the total of 221 subjects.

(if the IOTF criteria were used, 5% of 221= 11 underweight subjects, and if the underweight subjects were excluded, it means that normal weight, overweight and obese should be 210 (221-11=210), not 202 (156+31+15=202) as shown in table 2, so the numbers are not matching.

So please, correct all numbers in the tables and text, since they do not match with table 1 and with the text.

3.       There are some new parts which are inserted without point (see below, particularly lines 119-123, which are referring to randomization procedure for intervention, and in this article there is no intervention).

Even some referees asked for such changes (e.g., to add some more information in introduction), the inserted parts are too long, with too much details given, repetitive and more belong to discussion, than to introduction, which has to be very concise and short. Please, summarize them. (see below)

4.       I still think that Bonferroni correction for only 3 variables in regression models is not necessary. In turn, it must be applied when too many variables are added (e.g., 50 covariates tested in the other part of the manuscript).

Therefore, consider if you really need to correct p values in Concerning KIDMED questionnaire, more than 50.0% of the children needed to improve their diet and 7.50% had a very low-quality diet. The total score obtained in the KIDMED questionnaire was inversely associated with the schoolchildren’s BMI (β= -0.19 (95% IC -0.38-0), p = 0.04) and FM (β= -0.65 (-1.15--0.14), p=0,01). However, these associations lost significance after correction by multiple tests.”.

Was it necessary to lose this significance? I would not correct with Bonferroni only for 3 predictors.

5.       Additionally, I am not sure if the track changes in the text related to deletion are labeled (they are not separately visible)

For example: 27: “(twice repeated 24-hour48-hour food record)” should be “(twice repeated 24-hour food record)”

More detailed corrections in the text:

36-37: “A protective effect against EW depending on the number of dairy servings/day (OR=0.48 (0.29-0.75, p adjusted =0.05) was identified” – it should be more precise and concise in terms of statistics: “The number of dairy servings/day had a protective effect against EW (OR=0.48 (0.29-0.75, p adjusted =0.05).”

Additionally, I see some added sentences, for which I do not see the point, they are too long, repetitive, and do not give any extra values to the manuscript. The manuscript should be very concise, particularly introduction.

For example: 60-62: I do not see the point of the inserted sentences: “Prevention and treatment are two key determinants to assess in the context of childhood obesity. The development, implementation and evaluation of cost-effective prevention strategies are a high priority [8].”

It’s a repetition of the same point from the surrounding text and well known information, too general. Please, summarize or delete.

The same for: 86-88: “ The study of environmental factors concerning nutritional status is of great interest since they have indisputable importance in the pathogenesis of obesity and have been responsible for being able to define a classical Western society as "obesogenic" [18].” - Quite unclear, long and repetitive of the previous text. I would suggest to delete this sentence.

71-85: This introduced part is too long and looks more as a discussion, and should be summarized maximally:

“In Spain, very few studies have been carried out on eating habits in the pediatric population to fight against childhood obesity. Almost all of them are descriptive and cross-sectional studies. The ALADINO study [4] was carried out with a representative sample of the school population from 6 to 9 years of age, and the EsNuPi study [11] was carried out with a representative sample of Spanish children from 1 to 10 years of age residing in urban areas (except Ceuta and Melilla). There are the two most complete national studies on childhood age about their specific objectives related to the study of the prevalence of obesity and the study associated environmental factors. However, these studies do not include nutritional intervention. The THAO Child Health Program [12] and the POIBC study [13] were studies that collected data on nutritional status and lifestyle, with community nutritional intervention in Spain, but not carried out in the social context of the city of Madrid. Several studies have been conducted in other countries to study the association between nutritional and physical activity factors and excess weight in schoolchildren, underlining the need for more research in this area [14, 15, 16, 17].”

Please, summarize in only 2-3 sentences. For example: “Many studies have been conducted worldwide to study the association between nutritional and physical activity factors and excess weight in schoolchildren, underlining the need for more research in this area [14, 15, 16, 17, 18]. However, in Spain, only several studies were conducted, including two cross-sectional studies: the ALADINO study [4], examining school population 6-9 years old, and the EsNuPi study [11], examining children 1-10 years old residing in urban areas, and two interventional studies, the THAO Child Health Program [12] and the POIBC study [13], which also collected data on nutritional status and lifestyle. Nevertheless, having in mind …(please add information why your study is different)…., more studies in this area are needed”  

(i.e., add specific information why more studies are needed in this area in Spain, e.g., differences in the studied populations- region, age; and differences in methodology, etc.)

88: “Thus, the objective of the study was to examine the … in a group of schoolchildren (6–9 years old)…” (please, add this information)

119-123: Why this paragraph was inserted, when there is no intervention in the present study?? “For the nutritional intervention, randomization was carried out by school center. Thus, participating schools were randomly and proportionally stratified into 2 groups: intervention schools and control schools, considering the number of participants per center, their geographic area and their socioeconomic status. The randomization procedure was carried out with the statistical software R version 3.4 (www.r-project.org).

Please, delete this paragraph.

217-218:  “According to the WHO criteria, 32.2% of the students evaluated had EW (18.1% overweight, 14.1% obesity).”  - Add here also information on underweight subjects (were they excluded from Table 2?)

Table 2: please, indicate which criteria for obesity were used. Additionally, “Normal Weight”: please, check if they were actually  “Normal Weight and Underweight”

228- 229: “When the relationship between the %EB and schoolchildren´s nutritional status was evaluated, it was found an inverse and significant association (β= -1.49 229 (-1.9--1.07), p= 1.53E+05 (p<0.01)” Please, instead of “nutritional status” state “BMI” (more specific). (nutritional status can be 2-3 categories, e.g., “normal weight” and “excess weight”)

376-378: “ and in addition we are more confident in the positive test obtained, without the need of additional confirmatory experiments as those required with e.g. approaches to control the false discovery rate.” - please, delete this part, too long, confusing.

Instead, add information that no other confounding factors included (e.g., information on the presence of overweight/ obesity/ underweight in the family, since you pointed out in Introduction the genetic influence on obesity, so you did not asked for obesity in family, as possible confounder).

Author Response

We would like to thank the referees for their very helpful comments regarding the submission of our paper entitled " Dietary and physical activity habits as conditioning factors of nutritional status among children of GENYAL STUDY" (Manuscript ID: ijerph-2056551):

(Changes are marked up using the “Track Changes” in the revised ms):

(Our comments are in blue in this letter)

REVIEWER 1

I would like to thank the authors for detailed answers, they were quite satisfactory.

There are just several points which are not clear.

1. “Were there underweight kids in the cohort?

Yes, 5% of children with underweight were found in the sample following the IOTF criteria. Given the small sample size, they could not be treated independently.”

Were they excluded from further analyses, or were merged with “normal weight”? This information should be given in the text (lines 217-219).

Statistical analyzes between the study variables and nutritional status were performed with BMI as a continuous variable and as a BMI category according to IOTF criteria as a categorical variable, including underweight, average weight, and excess weight. Yes, 5% of children with underweight were found in the sample following the IOTF criteria. Given the small sample size, they could not be treated independently.

To clarify this aspect in the manuscript the following sentence has been introduced in the Statistical analyses section: (Ln 226-31): “Overweight and obesity categories were unified in some analyses as a single category called excess weight (EW). Statistical analyzes between the study variables and nutritional status were performed with BMI as a continuous variable and as a BMI category according to IOTF criteria as a categorical variable, including underweight, average weight, and overweight”, and in the results section (Ln 265-7): “Eleven underweight children (5%) were found in the sample following the IOTF criteria. Given the small sample size, they could not be treated independently.”

2. In Table 2, which criteria for definition of obesity/overweight were used? Please, specify in the table.

In the table 2, the numbers of normal weight, overweight and obese (156+31+15=202) do not match with the numbers (percentages) given in the text lines 217-219, nor with the total of 221 subjects.

(if the IOTF criteria were used, 5% of 221= 11 underweight subjects, and if the underweight subjects were excluded, it means that normal weight, overweight and obese should be 210 (221-11=210), not 202 (156+31+15=202) as shown in table 2, so the numbers are not matching.

So please, correct all numbers in the tables and text, since they do not match with table 1 and with the text.

We appreciate the reviewer seeing this error in the manuscript. We proceeded to review the data from all the analyzes, and we noticed that Table 2 contained errors in the n by category. We have changed the n per category following IOTF criteria in table 2 to keep the whole article with the same criteria (Ln 285). (221= 154 normal weight, 36 overweight, 20 obesity and 11 underweight).

According to the referee's suggestion, the Table 2 title was replaced (Ln 269): “Table 2. Main diet characteristics of the schoolchildren in accordance with nutritional status (IOTF criteria)”.

3. There are some new parts which are inserted without point (see below, particularly lines 119-123, which are referring to randomization procedure for intervention, and in this article there is no intervention).

We include the information on the stratification of the schools of this study according to the suggestion of reviewer 2.

Even some referees asked for such changes (e.g., to add some more information in introduction), the inserted parts are too long, with too much details given, repetitive and more belong to discussion, than to introduction, which has to be very concise and short. Please, summarize them. (see below)

According to the referee's suggestion we summarize the new information added in the introduction section (Ln 77-87):”Many studies have been conducted worldwide to study the association between nutritional and physical activity factors and excess weight in schoolchildren, underlining the need for more research in this area [11, 12, 13, 14, 15]. Nevertheless, in Spain, only several studies were conducted, such as The ALADINO study [4] and the EsNuPi study [16], the two most complete national studies on the prevalence of pediatric obesity, but they were descriptive and cross-sectional studies. Other national studies, such as The THAO Child Health Program [17] and the POIBC study [18], included nutritional intervention; but were carried out outside the social context of Madrid. The GENYAL study is a cluster randomized clinical trial with a 5-year follow- up intervention based on nutritional education, annual anthropometric measurement evaluations and data collection from questionnaires developed in Madrid (Spain), which will increase knowledge in this area”.

4. I still think that Bonferroni correction for only 3 variables in regression models is not necessary. In turn, it must be applied when too many variables are added (e.g., 50 covariates tested in the other part of the manuscript).

Therefore, consider if you really need to correct p values in “Concerning KIDMED questionnaire, more than 50.0% of the children needed to improve their diet and 7.50% had a very low-quality diet. The total score obtained in the KIDMED questionnaire was inversely associated with the schoolchildren’s BMI (β= -0.19 (95% IC -0.38-0), p = 0.04) and FM (β= -0.65 (-1.15--0.14), p=0,01). However, these associations lost significance after correction by multiple tests.”.

Was it necessary to lose this significance? I would not correct with Bonferroni only for 3 predictors.

In the first analysis (β= -0.19 (95% IC -0.38-0), p = 0.04) and FM (β= -0.65 (-1.15-- 0.14), p=0,01) the data were adjusted for sex and age. These associations lost significance after multiple comparison analysis that included a total of 50 variables from different categories included in the GENYAL study (nutritional, anthropometric, socio-sanitary, etc.).

To clarify this aspect in the manuscript, the authors removed some sentences referring to the loss of significance after multiple adjustments. They included a penalty at the end of the results section (Ln 475-6): “Only the relationship between the %EB and schoolchildren ́s BMI (p<0.01) and FM (p<0.01) maintained significance after multiple comparison analysis.”

5. Additionally, I am not sure if the track changes in the text related to deletion are labeled (they are not separately visible)

For example: 27: “(twice repeated 24-hour48-hour food record)” should be “(twice repeated 24-hour food record)”

The authors have corrected the sentence in the abstract (Ln 27): “(twice repeated 24-hour food record)”

More detailed corrections in the text:

36-37: “A protective effect against EW depending on the number of dairy servings/day (OR=0.48 (0.29-0.75, p adjusted =0.05) was identified” – it should be more precise and concise in terms of statistics: “The number of dairy servings/day had a protective effect against EW (OR=0.48 (0.29-0.75, p adjusted =0.05).”

According to the referee suggestion this phrase was replaced in the abstract (Ln 36-7): “The number of dairy servings/day had a protective effect against EW (OR=0.48 (0.29- 0.75, p adjusted =0.05)”

Additionally, I see some added sentences, for which I do not see the point, they are too long, repetitive, and do not give any extra values to the manuscript. The manuscript should be very concise, particularly introduction.

For example: 60-62: I do not see the point of the inserted sentences: “Prevention and treatment are two key determinants to assess in the context of childhood obesity. The development, implementation and evaluation of cost-effective prevention strategies are a high priority [8].”

According to the referee suggestion this phrase was replaced in the introduction (Ln 66- 8): “and can act as critical modulators in the prevention and treatment of obesity [8]”

It’s a repetition of the same point from the surrounding text and well known information, too general. Please, summarize or delete.

The same for: 86-88: “ The study of environmental factors concerning nutritional status is of great interest since they have indisputable importance in the pathogenesis of obesity and have been responsible for being able to define a classical Western society as "obesogenic" [18].” - Quite unclear, long and repetitive of the previous text. I would suggest to delete this sentence.

According to the referee suggestion this phrase was deleted.

71-85: This introduced part is too long and looks more as a discussion, and should be summarized maximally:

“In Spain, very few studies have been carried out on eating habits in the pediatric population to fight against childhood obesity. Almost all of them are descriptive and cross-sectional studies. The ALADINO study [4] was carried out with a representative sample of the school population from 6 to 9 years of age, and the EsNuPi study [11] was carried out with a representative sample of Spanish children from 1 to 10 years of age residing in urban areas (except Ceuta and Melilla). There are the two most complete national studies on childhood age about their specific objectives related to the study of the prevalence of obesity and the study associated environmental factors. However, these studies do not include nutritional intervention. The THAO Child Health Program [12] and the POIBC study [13] were studies that collected data on nutritional status and lifestyle, with community nutritional intervention in Spain, but not carried out in the social context of the city of Madrid. Several studies have been conducted in other countries to study the association between nutritional and physical activity factors and excess weight in schoolchildren, underlining the need for more research in this area [14, 15, 16, 17].”

Please, summarize in only 2-3 sentences. For example: “Many studies have been conducted worldwide to study the association between nutritional and physical activity factors and excess weight in schoolchildren, underlining the need for more research in this area [14, 15, 16, 17, 18]. However, in Spain, only several studies were conducted, including two cross-sectional studies: the ALADINO study [4], examining school population 6-9 years old, and the EsNuPi study [11], examining children 1-10 years old residing in urban areas, and two interventional studies, the THAO Child Health Program [12] and the POIBC study [13], which also collected data on nutritional status and lifestyle. Nevertheless, having in mind ...(please add information why your study is different)...., more studies in this area are needed”

(i.e., add specific information why more studies are needed in this area in Spain, e.g., differences in the studied populations- region, age; and differences in methodology, etc.)

According to the referee ́s suggestion this introduced part was replaced in the manuscript (Ln 77-87): “Many studies have been conducted worldwide to study the association between nutritional and physical activity factors and excess weight in schoolchildren, underlining the need for more research in this area [11, 12, 13, 14, 15]. Nevertheless, in Spain, only several studies were conducted, such as The ALADINO study [4] and the EsNuPi study [16], the two most complete national studies on the prevalence of pediatric obesity, but they were descriptive and cross-sectional studies. Other national studies, such as The THAO Child Health Program [17] and the POIBC study [18], included nutritional in-tervention; but were carried out outside the social context of Madrid. The GENYAL study is a cluster randomized clinical trial with a 5-year follow-up intervention based on nu-tritional education, annual anthropometric measurement evaluations and data collection from questionnaires developed in Madrid (Spain), which will increase knowledge in this area.”

88: “Thus, the objective of the study was to examine the ... in a group of schoolchildren (6–9 years old)...” (please, add this information)

According to the referee ́s suggestion the objective was replaced in the introduction section (Ln 88-91): “Thus, the objective of this study was to describe the main lifestyle characteristics (diet and exercise) and their possible association with nutritional status in a group of schoolchildren (6-9 years old) enrolled in the GENYAL study (Madrid, Spain) for the prevention of childhood obesity”.

119-123: Why this paragraph was inserted, when there is no intervention in the present study?? “For the nutritional intervention, randomization was carried out by school center. Thus, participating schools were randomly and proportionally stratified into 2 groups: intervention schools and control schools, considering the number of participants per center, their geographic area and their socioeconomic status. The randomization procedure was carried out with the statistical software R version 3.4 (www.r- project.org). “

Please, delete this paragraph.

Following the reviewer's suggestions, the authors have deleted this paragraph. Likewise, they state that it was added as a suggestion by reviewer 2, who validated his part of the corrections.

217-218: “According to the WHO criteria, 32.2% of the students evaluated had EW (18.1% overweight, 14.1% obesity).” - Add here also information on underweight subjects (were they excluded from Table 2?)

According to the referee ́s suggestion this information was added (Ln 265-267): “Eleven underweight children (5%) were found in the sample following the IOTF criteria. Given the small sample size, they could not be treated independently.”

Table 2: please, indicate which criteria for obesity were used. Additionally, “Normal Weight”: please, check if they were actually “Normal Weight and Underweight”

According to the referee ́s suggestion this information was added in the table ́s title (Ln 268): “Table 2. Main diet characteristics of the schoolchildren in accordance with nutritional status (IOTF criteria).”

228- 229: “When the relationship between the %EB and schoolchildren ́s nutritional status was evaluated, it was found an inverse and significant association (β= -1.49 229 (- 1.9--1.07), p= 1.53E+05 (p<0.01)” Please, instead of “nutritional status” state “BMI” (more specific). (nutritional status can be 2-3 categories, e.g., “normal weight” and “excess weight”).

The authors modified this part as a suggestion from previous reviews. Currently in the manuscript is how (Ln 273-6): “When the relationship between the %EB (independent variable) and schoolchildren ́s BMI and FM (dependent variables) was evaluated, it was found an inverse and significant association (β= -1.49 (-1.9--1.07), p= 1.53E+05 (p<0.01) and β= -3.46 (-4.62--2.29), p= 1.24E+08 (p<0.01), respectively)”

376-378: “ and in addition we are more confident in the positive test obtained, without the need of additional confirmatory experiments as those required with e.g. approaches to control the false discovery rate.” - please, delete this part, too long, confusing.

Following the reviewer's suggestions, the authors have deleted this part.

Instead, add information that no other confounding factors included (e.g., information on the presence of overweight/ obesity/ underweight in the family, since you pointed out in Introduction the genetic influence on obesity, so you did not asked for obesity in family, as possible confounder).

According to the referee ́s suggestion this information was added in the limitations section (discussion) (Ln 593-6): “The lack of other confounding factors, such as sociodemographic variables of the characteristics of the families or genetics variables, is considered a limitation of this study. Nevertheless, in the GENYAL study, these variables were collected and will be evaluated within other studies with framed objectives on this subject”.

The authors thank the reviewer for the comments in order to improve their manuscript.

Reviewer 2 Report

With the modifications made, the authors significantly improved the quality of the manuscript.

I am quite satisfied with the new version, congratulations on the good work.

Author Response

The authors thank the reviewer for the comments in order to improve their manuscript.

Reviewer 3 Report

The authors (AA) have carefully addressed the reviewers' comments. Overall the changes made have improved the manuscript.

Author Response

(The authors gave the same response as above.)
